# Common mental health problems and associated factors among recovered COVID-19 patients in rural area: A community-based survey in Bangladesh

Zobayer Ibne Zaid[1]⊚, Anika Tasnim[2]⊚, Md Maruf Haque Khan[2], Zubair Ahmed Ratan[3], Mohammad Tanvir Islam[4], M. Atiqul Haque[ID][2]*

1 Health Service Division, Ministry of Health and Family Welfare, Dhaka, Bangladesh, 2 Department of Public Health and Informatics, Bangabandhu Sheikh Mujib Medical University, Shahbag, Dhaka, Bangladesh, 3 Department of Biomedical Engineering, Khulna University of Engineering and Technology, Khulna, Bangladesh, 4 Department of Internal Medicine, Bangabandhu Sheikh Mujib Medical University, Shahbag, Dhaka, Bangladesh

⊚ These authors contributed equally to this work.
* atiqulm26@bsmmu.edu.bd

**Data Availability Statement:** Data are available from the Institutional Data Access / Ethics Committee (contact via Mr. Sobuj, Email:

## Abstract

### Purpose

Since the coronavirus (COVID-19) was announced as being a global pandemic on 11 March, governments from all parts of the world declared a quarantine period, during which people were prohibited from leaving their homes (except for essential activities) to contain the spread of the virus. Since then, the population has faced different levels of restrictions (i.e., mobility, social activities) that limited participation in normal daily routines. Consequently, these restrictions may have adversely changed physical activity, diet, sleep patterns, and screen time or work routine. So, the pandemic has had profound influence on the mental health of the entire societies. As the mental health status of Bangladeshi patients living in rural area that have recovered from COVID-19 has not been previously studied, this gap is addressed through the present investigation focusing on one rural Bangladeshi community.

### Methods

A convenience sampling method was employed to recruit participants for this cross-sectional study. Data was gathered by conducting face-to-face interviews with 243 recovered COVID-19 patients (as confirmed by a positive Reverse Transcription PCR test) attending a local primary health care facility center and instructed to consider how they felt in the preceding week

### Results

By administering a validated Bengali version of the Depression, Anxiety, and Stress Scale (DASS-21) to measure participants' mental health status, we noted that 24% of the sample

[exp2opi@gmail.com](mailto:exp2opi@gmail.com)) for researchers who meet the criteria for access to confidential data.

**Funding:** This study was partially funded by Bangabandhu Sheikh Mujib Medical University. The funders had no role in study design, data collection and analysis, decision to publish, or preparation of the manuscript. The authors received no specific funding for this work.

**Competing interests:** he authors have declared that no competing interests exist.

exhibited depressive symptoms. In addition, 30.9% and 21.8% of the participants experienced stress symptoms and reported anxiety, respectively. Sociodemographic factors such as female sex, lower educational level, living away from family, smaller living accommodations, and lower economic status significantly predicted mental health outcomes in multivariate logistic regressions.

## Conclusion

These results may help health care providers formulate proper mental health interventions and preventive measures to minimize the mental health problems among patients that have recovered from COVID-19.

## Introduction

The psychological impact of infectious disease pandemics on populations can have significant effects on disease transmission, emotional distress, and social dysfunction during and after the pandemic. The COVID-19 pandemic has resulted in physical and emotional health repercussions. While researchers have primarily focused on physical manifestations and complications, there is growing recognition of the mental health issues that may arise after COVID-19 recovery. A study conducted in China found that 54% of respondents rated the psychological effects of the pandemic as moderate or severe, with 29% reporting moderate to severe anxiety symptoms and 17% reporting moderate to severe depressive symptoms [1]. A systematic review found high rates of depression, anxiety, post-traumatic stress disorder (PTSD), psychological distress, and stress among the global population [2]. As the evidence on the pandemic's impacts continues to expand, it is becoming increasingly clear that COVID-19 has had a profound influence on the mental health of entire societies due to changes in daily routines [3]. Several studies suggest a link between the neuro-invasive potential of SARS-CoV-2 and the increase in mental illness incidence during COVID-19 infection and following recovery [4]. Data on the long-term psychological effects of COVID-19, along with other clinical complications, are accumulating as the pandemic enters its fourth year [5]. Given the growing body of evidence linking COVID-19 to mental health problems, it is essential to monitor patients who have recovered from COVID-19 after hospital discharge. Epidemiological statistics in Bangladesh have shown that the COVID-19 pandemic and widespread isolation have had a significant negative impact on mental health [6, 7]. For example, a survey of university students in Bangladesh found that 53% of participants had a moderate to poor psychological status, 40% had moderate to severe anxiety, and 72% had depressive symptoms [8]. Although some studies have been conducted to explore the mental health condition post-COVID-19 among the general population, including health workers and nurses [9, 10], no such study has specifically targeted rural communities.

Bangladesh is one of the most densely populated countries globally, with two-thirds of its 163 million residents living in rural areas, where there may be specific difficulties regarding stigma, access to mental health care, and cultural attitudes towards mental health. As a result, it is essential to evaluate the mental health status of the rural population post-COVID-19. This study aims to assess the mental health condition and associated factors of people recovering from COVID-19 in rural areas of Bangladesh. By identifying typical mental health issues and associated factors, this study can aid in the development of culturally appropriate interventions to assist the mental health and welfare of recovered COVID-19 patients in various regions.

The findings of this study may help us understand the mental health condition of people recovering from COVID-19 in low-income and rural areas, and the information can guide public health policies and initiatives to promote their mental health.

## Methods

### Setting and study population

This cross-sectional study was conducted at Sirajdikhan Upazila (sub-district) in the Munshiganj district of Bangladesh. Since the beginning of the COVID-19 outbreak in Bangladesh in March 2020, the Upazila Health Complex (UHC) of Sirajdikhan has been maintaining a database of Reverse Transcription PCR (RT-PCR) confirmed cases which was the basis of our study population. Until September 2020, 500 cases were reported at Sirajdikhan UHC. After excluding children and patients with active infection (within 14 days after diagnosis), 320 recovered COVID-19 patients remained, and we contacted each individual over the phone to invite them to take part in the study. We have not included the patients who take more than 14 days to recover after COVID 19. As, 243 of these individuals agreed to an interview, the non-response rate was 24%. All interviews were conducted in September 2020, during which it was confirmed that none of the participants had a known prior history of mental illnesses.

### Data collection

A semi-structured questionnaire was developed for use during the individual interviews. It probed into the sociodemographic information, such as sex, age, marital status, number of children, family size, position within the family (household head or other), living arrangements (living with a family member or alone during COVID-19), educational attainment, occupation (within last twelve months), monthly family income (USD equivalent), and accommodation size (number of rooms in the household). Medical information regarding BCG vaccination, any comorbidities, and blood group, was also collected. The mental health status of the participants was evaluated using the Bengali version of the Depression, Anxiety, and Stress Scale (DASS-21) [11]. It comprises of 21 items forming three self-reported subscales (with 7 items each) designed to assess depression, anxiety, and stress. All items require response on a 4-point Likert scale (0 = never a problem, 1 = sometimes a problem, 2 = often a problem, and 3 = almost always a problem) whereby the participants were instructed to consider how they felt in the preceding week. The final score for each subscale was multiplied by two and was evaluated according to its severity rating index. The scores for depression, anxiety, and stress were calculated by adding up the scores related to all items across the three subscales. The results are interpreted as described in the supporting information (S1 Table). In our study we have found the Cronbach alpha for depression, anxiety and stress was 0.82, 0.77 and 0.79 respectively whereas a study done on medical students in Bangladesh found it 0.987, 0.957, 0.964 respectively [11].

Five data collectors and one supervisor were responsible for data collection. Prior to commencing the interviews, the data collectors completed a five-day training program, allowing them to learn how to introduce themselves, explain the purpose of the study, obtain informed consent, and preserve confidentiality. Once the questionnaire was pre-tested among a similar population at a location outside the study area to ensure the ease of its interpretability, the face-to-face interviews were scheduled over the phone and were conducted at a place and time chosen by the participants.

### Ethics approval

This study was performed after obtaining ethical clearance from the Institutional Review Board (IRB) of Bangabandhu Sheikh Mujib Medical University (BSMMU) Shahbag, Dhaka,

Bangladesh (Reference No. BSMMU/2020/9624, Date: 07/11/2020). Informed consent was obtained from all recovered COVID-19 patients that were included in the study prior to collecting data. The consent form contains details about the study's aim and objectives, study procedure, benefit and risk of participation, right to refuse to participate or withdraw from the survey, confidential handling of data, and the responsible principal investigator's identity. The participants had the right to refuse the interviewer or withdraw him or herself from the study at any point during the study period.

### Statistical analysis

The data related to sociodemographic information, as well as participants' anxiety, stress, and depression scores, was subjected to descriptive analysis. Frequencies and percentages were calculated as summary measures for the qualitative variables, and arithmetic means and standard deviations (SDs) were used to describe the quantitative variables. The Kolmogorov–Smirnov test was conducted to determine whether continuous variables were normally distributed. To examine the associations of sociodemographic factors with the participants' mental health state the Chi Square test, independent sample t test was used for testing relationships between categorical variables and means of two groups respectively. Chi square test and independent sample t test was done to explore the prevalence of depression, anxiety, and stress in relation to different sociodemographic factors.

A logistic regression model was developed to identify the most relevant factors for predicting anxiety, stress, and depression. The final model was selected using the enter method, the coefficients' significance was established via Wald tests, and the goodness of fit of the model was assessed by the Hosmer–Lemeshow test. The results were described using odds ratio (OR) and strengths of associations were demonstrated by the Beta coefficient with a 95% confidence interval (CI), whereby *p-value* < .05 was considered statistically significant. The statistical software Statistical Package for the Social Sciences 26 (SPSS version 26 for Windows, IBM Corp., Chicago, IL, USA) was used to complete data codification, processing, and analysis.

## Results

Around 67% of the recovered COVID-19 patients were male and the mean (SD) age was 40.33 (12.88) years. Majority of the participants were married (81.1%) and had at least one child (76.1%). Most of the participants (88.5%) lived with their families, and 49.8% of the sample had more than four family members (Table 1).

When asked about prior inoculations, 86.4% of the respondents stated that they have had BCG vaccination, with BCG vaccination marks evident in 85.2% of the cases. The most prevalent blood group was B (ve+) (29.2%) followed by O (ve+) (24.7%) and A (ve+) (19.8%), while 13.2% of respondents were unaware of their blood group. Hypertension (28.8%) and diabetes (24.7%) were most prevalent comorbidities in our sample. We have included information related to comorbidities, BCG inoculation and blood type to determine any association of this to mental health status of recovered COVID-19 patients. But we have not found any statistical significance. We have considered depressive symptoms, stress symptoms and anxiety symptoms as mental health conditions. In response to questions about their mental health since recovering from COVID-19, we found that 40.7% of participants have a mental disorder (anxiety, depression, or stress). Among them 23.9% of the participants stated that they had been suffering from depressive symptoms, which were mostly mild (10.7%) and moderate (10.3%). Among the 21.8% of respondents that had symptoms of anxiety, 10.7% had mild, 5.8% had severe, and 2.9% had extremely severe anxiety symptoms. The greatest percentage (30.9%) of

**Table 1. Prevalence of depression, anxiety, and stress in relation to different sociodemographic and clinical factors (n = 243).**

| Variables | N = 243 (%) [#] | Depressive Symptoms, n (%) [$] | P value | Anxiety Symptoms, n (%) [$] | P value | Stress Symptoms, n (%) [$] | P value |
|---|---|---|---|---|---|---|---|
| **Sex** | | | | | | | |
| Female | 33.3 | 24 (29.6) | .152 | 29 (35.8) | 0.000*** | 34 (42.0) | .012** |
| Male | 66.7 | 34 (21.0) | | 24 (14.8) | | 41 (25.3) | |
| **Age (years) Mean (SD)** | 40.33(12.88) | 42.09 (14.10) | .086 | 38.92 (12.98) | .836 | 38.88 (12.67) | .968 |
| **Marital status** | | | | | | | |
| Married | 83.8 | 42 (21.3) | .082 | 36 (18.3) | .009*** | 53 (26.9) | .008*** |
| others | 16.2 | 16 (34.8) | | 17 (37) | | 22 (47.8) | |
| **Having children** | | | | | | | |
| No | 23.9 | 15 (25.9) | .725 | 14 (24.1) | .716 | 22 (37.9) | .195 |
| Yes | 76.1 | 43 (23.2) | | 39 (21.1) | | 53 (28.6) | |
| **Family size** | | | | | | | |
| ≤ 4 people | 50.2 | 32 (26.2) | .452 | 29 (23.8) | .535 | 38 (31.1) | 1.000 |
| > 4 people | 49.8 | 26 (21.5) | | 24 (19.8) | | 37 (30.6) | |
| **Number of rooms in the household** | | | | | | | |
| ≤ 2 | 54.7 | 41 (30.8) | .006*** | 36 (27.1) | .030*** | 52 (39.1) | .003*** |
| > 2 | 45.3 | 17 (15.5) | | 17 (15.5) | | 23 (20.9) | |
| **Living with family** | | | | | | | |
| No | 11.5 | 6 (21.4) | 1.000 | 12 (42.9) | .007*** | 12 (42.9) | .191 |
| Yes | 88.5 | 52 (24.2) | | 41 (19.1) | | 63 (29.3) | |
| **Position in the family** | | | | | | | |
| Family head | 51.4 | 27 (21.6) | .452 | 18 (14.4) | .005 | 29 (23.2) | .009 |
| Other | 48.6 | 31 (26.3) | | 35 (29.7) | | 46 (39.0) | |
| **Educational level** | | | | | | | |
| Up to primary | 14.8 | 14 (38.9) | .033** | 12 (33.3) | .081 | 14 (38.9) | .328 |
| Secondary and above | 85.2 | 44 (21.3) | | 41 (19.8) | | 61 (29.5) | |
| **Occupation** | | | | | | | |
| Unpaid | 18.5 | 13 (28.9) | .439 | 14 (31.1) | .110 | 17 (37.8) | .286 |
| paid | 81.5 | 45 (22.7) | | 39 (19.7) | | 58 (29.3) | |
| **Monthly family income** | | | | | | | |
| < 30,000 BDT | 67.1 | 50 (30.7) | 0.000*** | 38 (23.3) | ,509 | 56 (34.4) | .105 |
| 30,000 and above | 32.9 | 8 (10) | | 15 (18.8) | | 19 (23.8) | |
| **Vaccine (BCG)** | | | | | | | |
| No | 13.6 | 9 (27.3) | .662 | 6 (18.2) | .658 | 12 (36.4) | .543 |
| Yes | 86.4 | 49 (23.3) | | 47 (22.4) | | 63 (30.0) | |
| **Comorbidities** | | | | | | | |
| No | 50.6 | 27 (22.0) | .548 | 28 (22.8) | .757 | 41 (33.3) | .409 |
| Yes | 49.4 | 31 (25.8) | | 25 (20.8) | | 34 (28.3) | |

* 1 Bangladeshi Taka (BDT) equivalent to 87 US Dollar

**Others = separated, divorced, widowed, etc

[#] Column percentage

** p < .05

*** p < .01

[$] Figure within parenthesis denoted corresponding row percentage

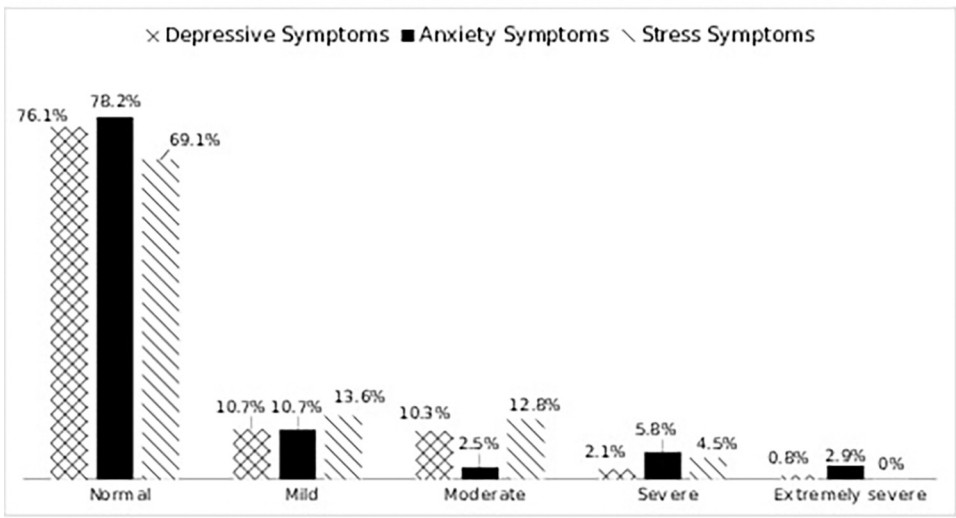

**Fig 1. The severity of depressive, anxiety, and stress symptoms among the recovered COVID-19 patients (n = 243).** The data are displayed in three distinct sections, each corresponding to a specific type of symptom experienced by the 243 individuals who have recovered from COVID-19: depressive symptoms, anxiety symptoms, and stress symptoms. For each symptom type, the severity is categorized into normal, mild, moderate, severe, and extremely severe.

the respondents experienced stress symptoms, including mild (13.6%), moderate (12.8%), and severe (4.5%) levels (Fig 1).

A considerably higher incidence of depressive (29.6%), anxiety (35.8%), and stress (42%) symptoms was noted among females compared to males. Higher levels of depression were also reported by unmarried compared to married individuals (54.5%). Depression was more prevalent among participants living in smaller accommodations (with 1–2 rooms) (30.8%), those with only primary-level education (38.9%), and with monthly income below 30,000 Bangladesh Taka (BDT) (30.7%). Depression accompanied by anxiety was most prominent among younger individuals (below the age of 23), while those in the 24–39 age group were affected by anxiety the most.

A Hosmer–Lemeshow test statistic of depressive symptoms ($\chi^2$ = 4.06, *p-value* = .85), anxiety symptoms ($\chi^2$ = 4.38, *p-value* = 0.82), and stress levels ($\chi^2$ = 4.18, *p-value* = .84) indicated that the model provided a good fit to the data because all significance values exceeded the .05 threshold. Multiple unadjusted logistic regression models also showed statistical significance. In particular, relative to females, male participants suffered less from anxiety (OR = 0.27; 95% CI = 0.09–0.81; *p-value* < .05). Relative to those with monthly family income below 30,000 BDT, those living on more than 30,000 BDT per month had a low odds ratio of exhibiting depressive symptoms (OR = .276; 95% CI = .112-.679; *p-value* < .01). In comparison to unmarried and others respondents, married participants have lower likely to experience anxiety (OR = .186; 95% CI = .053-.645; *p-value* < .01) and stress symptoms (OR = .351; 95% CI = .125-.983; *p-value* < .05) whereas those that lived in unsuitable accommodations) had a higher odds of experiencing high levels of stress(OR = 2.535; 95% CI = 1.283–5.008, *p-value* < .01) and depression(OR = 2.137; 95% CI = 1.023–4.467, *p-value* < .05). Furthermore, those who live with family have lower odds to experience anxiety OR = .290; 95% CI = .109-.769, *p-value* < .05) (Table 2).

**Table 2. Effects of socio-environmental factors on the mental health status of recovered COVID-19 patients (n = 243).**

| Variables | Depressive Symptoms | Anxiety Symptoms | Stress Symptoms |
|---|---|---|---|
| | Adjusted ORs (95%CI[a]) | Adjusted OR[b] s (95%CI) | Adjusted ORs (95%CI[a]) |
| **Sex** | | | |
| Female [ref] | | | |
| Male | .452 (.142–1.442) | .266(.088-.807) ** | .487(.177–1.343) |
| **Age (years)** | 1.023 (.986–1.061) | .986 (.947–1.027) | 1.000(.966–1.035) |
| **Marital status** | | | |
| others [ref] | | | |
| Married | .429 (.141–1.308) | .186(.053-.645) *** | .351(.125-.983) ** |
| **Having children** | | | |
| No [ref] | | | |
| Yes | 1.167 (.366–3.721) | 3.858(.965–15.422) | 1.589(.551–4.582) |
| **Family size** | | | |
| > 4 people [ref] | | | |
| ≤ 4 people | 1.146 (.573–2.290) | 1.290(.616–2.705) | .899(.475–1.701) |
| **Number of rooms in the household** | | | |
| > 2[ref] | | | |
| ≤ 2 | 2.137 (1.023–4.467) ** | 1.859(.855–4.040) | 2.535(1.283–5.008) *** |
| **Living with family** | | | |
| No [ref] | | | |
| Yes | 1.206 (.402–3.617) | .290(.109-.769) ** | .735(.292–1.848) |
| **Position in the family** | | | |
| Other [ref] | | | |
| Family head | .729 (.245–2.17) | .836(.283–2.470) | .640(.242–1.693) |
| **Educational level** | | | |
| Up to primary [ref] | | | |
| Secondary and above | .873 (.340–2.241) | .624(.218–1.789) | .879(.343–2.254) |
| **Occupation** | | | |
| Unpaid [ref] | | | |
| paid | 2.031 (.623–6.616) | 1.789(.572–5.592) | 1.534(.529–4.453) |
| **Monthly family income** | | | |
| < 30,000 BDT [ref] | | | |
| 30,000 and above | .276 (.112-.679) *** | .731(.317–1.686) | .649(.317–1.328) |
| **Vaccine (BCG)** | | | |
| No [ref] | | | |
| Yes | .770 (.307–1.933) | 1.653(.539–5.067) | .808(.344–1.899) |
| **Comorbidities** | | | |
| No [ref] | | | |
| Yes | 1.176 (.536–2.580) | 1.197(.525–2.730) | 1.055(.519–2.144) |
| Nagelkerke R Square | .175 | .222 | .156 |
| Cox & Snell R Square | .117 | .144 | .111 |

*Note*: ** $p < .05$

***$p < .01$

[a]95% CI: 95% confidence interval

[b]adjusted OR: adjusted Odds Ratios

## Discussion

The aim of this study was to assess the mental health status among recovered COVID-19 patients living in a rural community of Bangladesh, and our results indicate that mental health symptoms are highly prevalent (40.7% of the sample) in the studied area. Given that a recent national survey revealed around 18% prevalence of mental disorders among the adult population in Bangladesh, it is evident that COVID-19 has exacerbated this problem [12]. This is not surprising, as in addition to worries about getting seriously ill and potentially dying, most individuals have experienced lockdowns and significant changes in their daily routines. The resulting social isolation, pessimism, and social and financial insecurity might have led to anxiety, depressive symptoms, and stress among the participants [13].

Extant research indicates that infectious diseases can trigger psychiatric disorders [14]. For example, it is estimated that 10–18% of SARS survivors in Hong Kong had anxiety and depression [15] whereas 42.9% prevalence of PTSD was noted among MERS survivors in South Korea [16]. These results are comparable to the percentages of depressive stress symptoms (30.9%), depressive symptoms (23.7%) and anxiety (21.8%) in our COVID-19 recovered cohort. Our results are also comparable to those reported for discharged Chinese COVID-19 patients, 23% of whom reported depressive symptoms and 21.8% had mild to extreme anxiety symptoms [17].

Sex- and gender-related differences in the prevalence of mental health are frequently reported [18] irrespective of cultural setting or level of development in the country, with greater preponderance of women among individuals suffering from anxiety or mood disorders [19]. Our female participants also reported greater anxiety and stress levels relative to males, which concurs with the prevalence of mental health disorders in Bangladesh [20]. These findings can be attributed to the patriarchal society, as in Bangladesh "dominant male model" still prevails, especially in rural communities, whereby females are given little autonomy and are mainly responsible for the household chores and taking care of children [21].

Therefore, having to fulfill these responsibilities after recovering from COVID-19 likely played a role in the high rates of mental health issues among female participants.

We also found that depressive and anxiety symptoms were more likely to be reported by younger participants (aged below 23 years), which concurs with the previous finding that stress levels tend to decline with increasing age [22]. While no definitive conclusions can be reached regarding these disparities, it is possible that younger individuals have had much greater disruption to their lifestyle due to COVID-19 and, as they were previously in better health than older individuals, they found the recovery and the remaining COVID-19 effects on their health more difficult to accept. These arguments are supported by the results reported by other authors, as tertiary educational institutions in Bangladesh have been closed for a long period due to COVID-19 [23], leading to a considerable isolation among younger individuals, while causing uncertainty regarding their future educational path and career prospects, which might have provoked mental illness.

It is also worth noting that in extant studies, COVID-19 patients that had other underlying conditions, such as cardiovascular disease, hypertension, diabetes, respiratory disease, and other chronic disorders, were more likely to suffer from mental health issues [24].

However, in our study, no such relationship emerged, potentially due to a small sample size and only 120 individuals reporting presence of any comorbidities.

It is, however, encouraging to find that living with family seemed to serve as a protective factor against anxiety (OR = .290; 95% CI = .109-.769, *p-value* < .05), as COVID-19 recovery is a challenging process and without the emotional, behavioral, and practical support of loved ones can lead to psychological distress and loneliness [25].

Linked to the above, we found that married individuals also suffered less from stress compared to divorced, separated or widowed participants concurring with the results reported for medical workers in Ningbo, China (in whom stress was linked to insomnia, which was further related to marital status) [26]. These results can be explained by lower cortisol (a stress hormone) levels in married people [27], suggesting that close intimate relationships (i.e., marriage or cohabitation) might play an instrumental role in reducing stress levels during times of crisis, including the COVID-19 pandemic.

In line with the findings reported by other authors, educational status did not seem to correlate with the mental health issues in our cohort [28]. Nonetheless, participants who completed graduate-level education experienced more depression and anxiety, likely due to their greater awareness of the potential long-term consequences of COVID-19 [29]. Conversely, those living on lower income (< 30,000 Bangladesh Taka [BDT] per month; [1 US Dollar equivalent to 87 BDT]) experienced higher levels of depressive symptoms (OR = .276; 95% CI = .112-.679; *p-value* < .01) compared to those with a family income above 30,000 BDT. This finding is expected, given that financial hardship and economic uncertainty have been found to amplify mental illness [30].

## Limitations of the study

When interpreting the findings reported here, certain limitations implicit in this study design should be noted. Specifically, as all information used in analyses was self-reported, there is a considerable risk of recall bias. In addition, the sample was relatively small and comprised solely of individuals recruited from one upazila, due to which they cannot be generalized to other parts of Bangladesh. Further, due to the study design, no causal inferences could be established between any studied factors and mental health symptoms. A significant limitation of this study is the substantial non-response rate of 24%, despite the utilization of convenience sampling, which could be attributed to the prevailing stigma associated with mental health issues among people in Bangladesh. This cultural stigma might have made individuals apprehensive about disclosing their mental health-related information. Additionally, the period during which the study was conducted coincided with a relatively high death rate, possibly stemming from various health-related issues and external circumstances. Furthermore, the imposition of lockdown measures, possibly due to the prevailing circumstances, could have further hindered individuals' willingness and ability to participate in our study. Finally, as several confounding factors (such as government lockdown directives, home quarantine, scarcity of oxygen, lack of necessary drugs, financial insecurity, domestic violence, and exposure to media) were not considered in our model, these issues should be addressed in future investigations.

## Conclusion

This community-based survey in rural areas of Bangladesh highlights the mental health issues among recovered COVID-19 patients. The study found that a considerable percentage of respondents experienced anxiety, depression, and post-traumatic stress disorder. The study also identified several factors associated with the occurrence of mental health issues, including older age, female gender, congested accommodations, absence of family members, and lower levels of education and income. The findings emphasize the need for policymakers and healthcare providers to prioritize mental health services and support for recovered COVID-19 patients. Community-based strategies such as mental health education, awareness campaigns, and counseling and support services can be effective interventions. Incorporating remote mental health services, such as teletherapy and virtual support groups, is a promising avenue to

reach a wider demographic of recovered patients, particularly those dwelling in remote or underserved enclaves. Additionally, integrating mental healthcare into primary care and educating the public about mental health and stigma can have a significant impact on the mental health of the recovered COVID-19 patients. Given the socio-cultural context and existing policies and services in Bangladesh, the implementation of these strategies needs to be carefully planned and executed. Collaboration with international health organizations and experts could provide invaluable insights and resources to bolster mental health. Overall, this study provides valuable insights into the mental health status of recovered COVID-19 patients in rural areas of Bangladesh and can guide the development of appropriate interventions and policies to support their mental health and well-being.

## Supporting information

**S1 Table. DASS 21 subscales for scores for depression, anxiety, and stress.**
(DOCX)

## Acknowledgments

The authors would like to express their gratitude to Marium Salwa, Research Associate, Department of Public Health and Informatics, Bangabandhu Sheikh Mujib Medical University, Dhaka, Bangladesh. Her invaluable support played a crucial role in conducting this study. Additionally, the authors extend their heartfelt thanks to all the participants who took part in this research.

## Author Contributions

**Conceptualization:** Zobayer Ibne Zaid, Md Maruf Haque Khan, Mohammad Tanvir Islam, M. Atiqul Haque.

**Data curation:** Zobayer Ibne Zaid, Mohammad Tanvir Islam, M. Atiqul Haque.

**Formal analysis:** Anika Tasnim, Md Maruf Haque Khan, M. Atiqul Haque.

**Funding acquisition:** M. Atiqul Haque.

**Investigation:** Zobayer Ibne Zaid, Md Maruf Haque Khan, M. Atiqul Haque.

**Methodology:** Zobayer Ibne Zaid, Anika Tasnim, Md Maruf Haque Khan, Mohammad Tanvir Islam, M. Atiqul Haque.

**Project administration:** Zobayer Ibne Zaid, Md Maruf Haque Khan, M. Atiqul Haque.

**Resources:** M. Atiqul Haque.

**Software:** M. Atiqul Haque.

**Supervision:** Anika Tasnim, M. Atiqul Haque.

**Validation:** M. Atiqul Haque.

**Visualization:** Anika Tasnim.

**Writing – original draft:** Zobayer Ibne Zaid, Anika Tasnim.

**Writing – review & editing:** Anika Tasnim, Zubair Ahmed Ratan, Mohammad Tanvir Islam.

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
