## [Decision Letter · Decision Letter 0]

8 Sep 2022

PONE-D-22-13763Mental health status of patients that have recovered from COVID-19 in a selected rural community of Bangladesh: A cross-sectional studyPLOS ONE

Dear Dr. Haque,

Thank you for submitting your manuscript to PLOS ONE. After careful consideration, we feel that it has merit but does not fully meet PLOS ONE’s publication criteria as it currently stands. Therefore, we invite you to submit a revised version of the manuscript that addresses the points raised during the review process.

We look forward to receiving your revised manuscript.

Kind regards,

Eshetu Girma Kidane

Academic Editor

PLOS ONE

Journal Requirements:

Reviewers' comments:

Reviewer's Responses to Questions

**Comments to the Author**

1. Is the manuscript technically sound, and do the data support the conclusions?

Reviewer #1: Partly

Reviewer #2: Partly

2. Has the statistical analysis been performed appropriately and rigorously? 

Reviewer #1: Yes

Reviewer #2: I Don't Know

3. Have the authors made all data underlying the findings in their manuscript fully available?

Reviewer #1: Yes

Reviewer #2: No

4. Is the manuscript presented in an intelligible fashion and written in standard English?

Reviewer #1: Yes

Reviewer #2: No

5. Review Comments to the Author

Reviewer #1: I would like to thank for the opportunity given to review this article. Bllow are my comments and questions

Title

Metal health is broad concept and the instrument they used to address the concept mainly is for anxiety, depression, and stress. Having this thought in mind I will frame the title as Common mental disorders rather than mental health.

Introduction

Page 10 line 70-74 … you mentioned a previous study showing higher prevalence of MH among people recovered from COVID in Bangladesh. If there is a previous study showing the problem, then what is new in this study?

All in all, I am not well convinced in the introduction. The main part of this study is MH, however, most of the paragraphs in the study are about pandemics before COVID (which I don’t like). I will recommend reframing this section focusing on only COVID and Mental health and showing what new things that this study is going to address.

Methods

Methodologically I haven’t seen any mechanism that showed the effect of COVID on MH of the participants. I will have a control group without COVID so that I will have control over the effect of COVID alone. The prevalence reported here may not attributed to COVID only.

On the statistical analysis section the specific analysis technique used in multivariable level analyse was not mention (they only said in the final model we do …..). They also used in appropriate terminologies that does not fit to the objective of the study like multivariate analysis. For me, multivariate analyses are those analysis which involve interplay of multiple variables such as Factor analysis. I recommend revising this section based on the points I raised here. Furthermore, they said they have a cut-off value of <0.2 to decide variable for multivariable analyse. It looks that the researchers are looking for association blindly (without having a prior hypothesis). For me, variables to be included into the final model should be theory driven rather than statistic driven.

Result

Table 1: it will be good if the level of significance in terms of p-values will be added as one column (as this stand it gave little information)

Figure 1: the title should be below the figure and also I will use no depression rather than normal

The format of Table is not attractive, either reduce information or the font size of the table so that it will be attractive for readers.

Discussion

The prevalence of mental health reported is about 40% which is considered very high, this might be due to COVID but as I already said in the methodology section the effect of the COVID is not well controlled. Therefore, it is difficult to infer it to the pandemic (I don’t agree with the discussion). Finings such as higher prevalence among Female, and young participants and protective effect of good support system (family support, marital status, educational, and economic status), support the above point that in general population CMD is more prevalent in female and youngsters and lower odds in those who had good social support (still not convinced to attribute it to COVID)

Conclusion

It is not advised to use terms like risk in cross-sectional study. Rather use terms like increased odds and like.

Reviewer #2: First two sentences in the abstract are broad/general with no clear link to mental health. Further in the introduction section, line 49-51 – “As the evidence on its impacts continues to expand, it is becoming increasingly evident that the pandemic has had profound influence on the psychological health of the entire societies due to the changes in daily routines [3]. These mental health issues have been given greater attention in high-income countries” - The first statement is too crude – how the pandemic would have ‘psychological health’ on the entire society; what changes of daily routines should be clarified. The inconsistencies on ‘psychological health’ and ‘mental health issues’ is also worrying. What is the argument here?

Line 53-56 “…the severity of mental health problems in these parts of the world would have even greater detrimental consequences, given that in most LMICs the resources designated for mental health are limited (and can be below 1% of the health budget)” – Although this may be logically true, I am wondering if there are more recent evidences than the 2017 publications referred here?

The objective of the study is not clear – what is the intension of the manuscript? What is intended other than bridging the ‘gap’ which in itself is not clearer in terms of what gap it refers to.

Line 79-81 in the intro states – “Thus, there is a paucity of data that reveal the prevalence of depression, anxiety, and stress among Bangladeshi patients that have recovered from COVID-19” – Does this make sense when the title is about the status of mental health status of … So, it is important to spell out what objective is intended to be addressed.

Method – Selecting participants - how long after recovery? This may have good implication on ‘metal health issues’

Line 89-92 – “After excluding children and patients with active infection (within 14 days after diagnosis), 320 recovered COVID-19 patients remained, and we contacted each individual over the phone to invite them to take part in the study. As 243 of these individuals agreed to an interview, the non-response rate was 24%.”

Firstly, what if the patient did not recover after 14 days for there are those with COVID who may take extra days to get cleared? Secondly, how could the high nonresponse rate could affect your result – not indicated in the limitation section.

I am wondering if the Bengali version of the Depression, Anxiety, and Stress Scale (DASS-21) was checked if it indeed measures what it is supposed to measure? Am wondering if cronbach alpha was applied to that effect or how else was tool validated for this particular study? Mention was made on line 107 “The validated Bengali version was thus used in this study” while it is not clear on who validated it and by whom – no reference was provided.

Line 148-149 – “To examine the associations of sociodemographic factors with the participants’ mental health state, Chi-squared, Mann–Whitney, or t-tests were performed, as appropriate” – Which test was applied under what circumstance need to be clearly pointed out. “When appropriate’ may affect the assessment of relevance of test.

The statistical methods applied in this study especially the logistic regression as indicated on lines 150-153 do not seem to fit at least to the title which is about the status of mental health instead of associated factors nor do we have this clearer in the objective section. There is no reason/justification for why prediction was considered.

Result section did not clarify if depressive symptoms, stress symptoms and anxiety are considered as long term ‘mental health issues’. While the standard tool may have defined as such, this has to be well articulated here and perhaps later in the introduction.

What is the association between some (selected??) co-morbidities, BCG inoculation and blood type was not clear and association of these to the objective of the study? This was not clearer?

Discussion is fairly ok although with clearer objectives and subsequent restructuring of the result section, discussion section may improve.

The conclusion section gives an impression that in Bangladesh, psychological health is not given sufficient attention, improvement of mental health services need to improve… that do not have backing from the result. This needs to be re-written.

6. PLOS authors have the option to publish the peer review history of their article (what does this mean?). If published, this will include your full peer review and any attached files.

Reviewer #1: No

Reviewer #2: **Yes: **Mirgissa Kaba

---

## [Author Response · Author response to Decision Letter 0]

13 Apr 2023

response to reviwers is uploaded in attached file section

---

## [Decision Letter · Decision Letter 1]

21 Aug 2023

PONE-D-22-13763R1Common mental health problems and associated factors among recovered COVID-19 patients in rural area: A community-based survey in BangladeshPLOS ONE

Dear Dr. Haque,

Thank you for submitting your manuscript to PLOS ONE. After careful consideration, we feel that it has merit but does not fully meet PLOS ONE’s publication criteria as it currently stands. Therefore, we invite you to submit a revised version of the manuscript that addresses the points raised during the review process.

We look forward to receiving your revised manuscript.

Kind regards,

Hubert Amu, PhD

Academic Editor

PLOS ONE

Journal Requirements:

Additional Editor Comments (if provided):

Reviewers' comments:

Reviewer's Responses to Questions

**Comments to the Author**

1. If the authors have adequately addressed your comments raised in a previous round of review and you feel that this manuscript is now acceptable for publication, you may indicate that here to bypass the “Comments to the Author” section, enter your conflict of interest statement in the “Confidential to Editor” section, and submit your "Accept" recommendation.

Reviewer #2: All comments have been addressed

Reviewer #3: All comments have been addressed

2. Is the manuscript technically sound, and do the data support the conclusions?

Reviewer #2: Yes

Reviewer #3: Yes

3. Has the statistical analysis been performed appropriately and rigorously? 

Reviewer #2: Yes

Reviewer #3: Yes

4. Have the authors made all data underlying the findings in their manuscript fully available?

Reviewer #2: Yes

Reviewer #3: Yes

5. Is the manuscript presented in an intelligible fashion and written in standard English?

Reviewer #2: Yes

Reviewer #3: Yes

6. Review Comments to the Author

Reviewer #2: Good the comments were addressed. However, there are two queries that may have to be addressed:

a) findings show that 24%, 21% and 22% have exhibited depressive symptom, stress and anxiety respectively. Given there is no evidence on when after recovery was the interview made, this finding could probably suffer much way beyond the recall bias. What if they do not even remember of how they felt about those then?

b) 24% non-response is quite large a number despite consideration of convivence sampling. This warrants proper explanation and proper documentation in the limitation section

Reviewer #3: This is a good research; however, based on this study, the author can include several valuable recommendations to ensure that they can make the most out of this research. Such suggestions include:

Remote Mental Health Services: Given the challenges posed by the pandemic, remote mental health services like teletherapy and virtual support groups could play a significant role in reaching a broader population of recovered patients, particularly those in remote or underserved areas.

Collaboration with Global Health Organizations: Partnering with international health organizations and experts could offer valuable insights and resources for enhancing mental health services in Bangladesh. Sharing best practices and lessons learned from other countries can contribute to more effective interventions.

Policy Reforms and Funding Allocation: The urgency of addressing the mental health crisis should prompt policymakers to reevaluate resource allocation and policy reforms to prioritize mental health services. Adequate funding and policy support are essential for establishing a sustainable and effective mental health infrastructure.

7. PLOS authors have the option to publish the peer review history of their article (what does this mean?). If published, this will include your full peer review and any attached files.

Reviewer #2: **Yes: **Associate Prof Mirgissa Kaba

Reviewer #3: No

---

## [Author Response · Author response to Decision Letter 1]

31 Aug 2023

I have uploaded responds to reviewers version 2 in a seprate file and also submitted manuscript in track chnage and without track change

---

## [Decision Letter · Decision Letter 2]

20 Sep 2023

PONE-D-22-13763R2Common mental health problems and associated factors among recovered COVID-19 patients in rural area: A community-based survey in BangladeshPLOS ONE

Dear Dr. Haque,

Thank you for submitting your manuscript to PLOS ONE. After careful consideration, we feel that it has merit but does not fully meet PLOS ONE’s publication criteria as it currently stands. Therefore, we invite you to submit a revised version of the manuscript that addresses the points raised during the review process.

We look forward to receiving your revised manuscript.

Kind regards,

Hubert Amu

Academic Editor

PLOS ONE

Additional Editor Comments :

I invite the authors to revise the manuscript by paying critical attention to comments made by our cherished reviewers.

Reviewers' comments:

Reviewer's Responses to Questions

**Comments to the Author**

1. If the authors have adequately addressed your comments raised in a previous round of review and you feel that this manuscript is now acceptable for publication, you may indicate that here to bypass the “Comments to the Author” section, enter your conflict of interest statement in the “Confidential to Editor” section, and submit your "Accept" recommendation.

Reviewer #2: All comments have been addressed

Reviewer #3: All comments have been addressed

2. Is the manuscript technically sound, and do the data support the conclusions?

Reviewer #2: Yes

Reviewer #3: Yes

3. Has the statistical analysis been performed appropriately and rigorously? 

Reviewer #2: Yes

Reviewer #3: Yes

4. Have the authors made all data underlying the findings in their manuscript fully available?

Reviewer #2: Yes

Reviewer #3: Yes

5. Is the manuscript presented in an intelligible fashion and written in standard English?

Reviewer #2: Yes

Reviewer #3: Yes

6. Review Comments to the Author

Reviewer #2: Authors made substantive revisions where all the comments and queries were addressed.

Reviewer #3: I have carefully reviewed your research, and I must commend you and your team for the meticulous work you've done. It's evident that you have addressed all the recommendations presented in your research, showcasing a high level of dedication and expertise in the field.

Your thoroughness in implementing these recommendations not only strengthens the validity of your research but also underscores your commitment to producing high-quality work. This approach reflects positively on your professionalism and the credibility of your findings.

I believe your research sets an excellent example for others in the field, emphasizing the importance of addressing recommendations rigorously. It will undoubtedly contribute significantly to our understanding of the subject matter. Well done!

7. PLOS authors have the option to publish the peer review history of their article (what does this mean?). If published, this will include your full peer review and any attached files.

Reviewer #2: **Yes: **Mirgissa Kaba Serbessa

Reviewer #3: **Yes: **Farah Adilla Ab Rahman

---

## [Author Response · Author response to Decision Letter 2]

6 Oct 2023

A new Response to reviewers letter is updated as per the instruction

---

## [Editor Report · Decision Letter 3]

3 Nov 2023

Common mental health problems and associated factors among recovered COVID-19 patients in rural area: A community-based survey in Bangladesh

PONE-D-22-13763R3

Dear Dr. Haque,

We’re pleased to inform you that your manuscript has been judged scientifically suitable for publication and will be formally accepted for publication once it meets all outstanding technical requirements.

Kind regards,

Hubert Amu, PhD

Academic Editor

PLOS ONE

Additional Editor Comments (optional):

The authors have addressed comments made by the reviews who in their own assessments feel the manuscript is ready for publication. I, therefore, recommend acceptance of the manuscript.
---

## [Editor Report · Acceptance letter]

24 Mar 2024

PONE-D-22-13763R3 

PLOS ONE

Dear Dr. Haque, 

I'm pleased to inform you that your manuscript has been deemed suitable for publication in PLOS ONE. Congratulations! Your manuscript is now being handed over to our production team.

Kind regards, 

on behalf of

Dr. Hubert Amu 

Academic Editor

PLOS ONE